# Dynamic 5-Hydroxymethylcytosine Change: Implication for Aging of Non-Human Primate Brain

**Xiaodong Liu, Xiao-Jiang Li and Li Lin ***

Guangdong Key Laboratory of Non-Human Primate Research, Guangdong-Hongkong-Macau Institute of CNS Regeneration, Jinan University, Guangzhou 510632, China
* Correspondence: linli@jnu.edu.cn

**Abstract:** Profiling of 5-hydroxymethylcytosine (5hmC) in the brain regions of rhesus monkey at different ages reveals accumulation and tissue-specific patterns of 5hmC with aging. Region-specific differentially hydroxymethylated regions (DhMRs) are involved in neuronal functions and signal transduction. These data suggest that 5hmC may be a key regulator of gene transcription in neurodevelopment and thus a potential candidate for the epigenetic clock. Importantly, non-human primates are the ideal animal models for investigation of human aging and diseases not only because they are more genetically similar to humans but also epigenetically.

**Keywords:** non-human primates; epigenetic; brain region; aging; 5-hydroxymethylcytosine (5hmC)

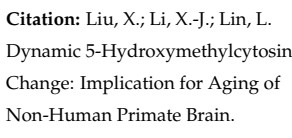

DNA methylation is essential for regulating gene expression in physiological processes of brain development and is critical for the occurrence and development of brain diseases. 5-methylcytosine (5mC), a classical form of DNA-methylation modification, has been recognized for regulating tissue- and cell-type-specific gene expression. As the first oxidized form of 5mC, 5-hydroxymethylcytosine (5hmC) has attracted great attention in recent years. Initially, 5hmC was considered to be a transient demethylation intermediate. Recent studies have shown a specifically higher distribution of 5hmC in the central nervous system, suggesting that 5hmC may be a key molecular marker for neurodevelopment and disease-related processes in the brain. Aging is a complex biological process modulated by multiple intrinsic and extrinsic factors, including epigenetic modifications [1]. Horvath et al. first proposed that the degree of 5mC at a specific site in DNA can be used to predict biological age and 5mC is now known as the epigenetic clock [2]. It has been reported that the major trend of aging includes global hypomethylation [3], which mainly occurs at repetitive DNA sequences. For a comprehensive understanding of the role of DNA methylation in development and aging, 5hmC should be considered.

To explore the mechanism of aging, animal models, including *Caenorhabditis elegans* and rodents, have been popularly used. However, these models are far from humans in terms of genetic and physiological similarities; in particular, their short life span is hard to compare with humans. More importantly, small animals lack the folding of the cortical surface, which is a unique structure in large mammals. Species-dependent differences in epigenetic regulation have been reported. For example, several studies demonstrated that pharmacological treatments could impact the epigenetic age to improve health or elongate lifespan in mice [4,5]; however, rapamycin did not significantly have an influence on the epigenetic age of the marmoset blood, though it could extend lifespan in mice [6]. DNA methylation studies of human aging were mainly performed using whole-blood samples because it is difficult to obtain fresh human-brain tissues. There were few studies of epigenetic regulation in postmortem human brains; however, the quality of post-mortem materials can be affected by the collecting times and conditions [7], which may introduce extensive variations considering that DNA methylation is sensitive to environmental changes. Cross-species conservation analyses demonstrated that the human-marmoset

age clock has a moderately high correlation to two other non-human primate species: vervet monkeys and rhesus monkeys [7]. These findings indicated that non-human primates are closer to humans in epigenetics and would be an ideal model for investigating aging-related epigenetics.

Thus, we used freshly collected and well-preserved rhesus monkey brain tissues to examine the genome distribution and dynamics of 5hmC in different brain regions at 2 (juvenile), 8 (young adult), and 17 (old) years of age [8]. We observed an overall accumulation of 5hmC with age in all four brain regions in monkeys. Previous studies have shown that profiling of 5hmC in mouse brain displayed an age-related accumulation of 5hmC in the cerebellum and hippocampus [9]. Analyses of frozen human tissue have revealed the increased 5hmC in senescent cells [10]. Consistent with these early findings, elevated 5hmC in the monkey brain during aging indicates that an age-dependent increase in 5hmC is conserved across mammalian species and that 5hmC could be a new candidate of the epigenetic clock. A comparison of the 5hmC patterns in the cerebellum of human, mice, and rhesus monkey revealed that the 5hmC feature of rhesus monkey is closer to that in human [8], consistent with the notion that rhesus monkey shares a high degree of similarity (93%) in the genetic homology to human. These findings further demonstrate that the non-human primate is a better animal model to investigate human aging and diseases.

Tissue-specific gene expression provides a fundamental biological framework for differentiated phenotypes and functions among tissues. In the context of the same genome, dynamic DNA-methylation modifications are critical for regulating the differential expression of genes. Analyzing 5hmC in different brain regions of rhesus monkey brain indicated a unique pattern of 5hmC modification in the cerebellum at all ages, while the striatum demonstrated specific 5hmC alterations in older monkeys [8]. A further analysis demonstrated that brain region-specific differentially hydroxymethylated regions (DhMRs) are enriched in neuronal function and signal-transduction pathways [8]. These data suggest that region- and age-dependent 5hmC is supposedly involved in the regulation of tissue-specific gene expression, which could also play a role in the pathogenesis of aging-related brain diseases. It is necessary to further analyze the relationship between the 5hmC change and gene expression of specific tissues during aging.

Meanwhile, senescence is accompanied by changes in gene expression; thereby, tissue-specific gene expression can induce senescence at different rates in different tissues. It has been demonstrated that the cerebellum ages more slowly than other parts of the body in human and senescence of the mouse cerebellum is earlier than the hippocampus [11,12]. The cerebellum-specific pattern 5hmC and DhMRs associated with neuronal function in monkey, might provide a new biomarker for the aging rate of the cerebellum. The functions of the cerebellum are not only involved in coordination and movement but are also related to the cognitive dysfunction associated with neurodegenerative diseases with ataxic symptoms [13,14]. Cerebellum-specific 5hmC patterns may also provide novel insights into the pathogenesis of cerebellar-associated neurodegenerative diseases. In addition, the age-dependent 5hmC distribution in the striatum was correlated with DhMR in the old monkey [8], which is enriched in learning and locomotory behavior pathways that are also associated with striatum-related neurodegeneration [15].

Despite the progress in the past few years in understanding the function of 5hmC [16,17], the exact role of 5hmC in the regulation of gene expression and aging process is still not clear. The dynamics of 5hmC in different brain regions of non-human primate provide a new insight into the epigenetic clock. Age-dependent and region-specific 5hmC can serve as a candidate biomarker for aging to help predict accurate chronological age. To comprehensively explore the contribution of DNA methylation to gene expression, development, and aging, 5mC should be considered together because 5mC is the substrate of 5hmC. Both 5mC and 5hmC play an integrative role on gene regulation. Schlosberg et al. reported that stable levels of 5mC plus 5hmC are associated with the repression of gene [18] and, consistently, decreases in 5mC plus 5hmC can activate gene expression [18]. Future studies should be performed to investigate the change of 5mC and transcription alterations

in different brain regions of rhesus monkey during aging. Integrated analyses of 5mC, 5hmC, and transcription will provide important and mechanistic insight into the DNA methylation in neurodevelopment and diseases.

**Author Contributions:** Writing—original draft preparation, X.L.; writing—review and editing, L.L.; supervision, X.-J.L. All authors have read and agreed to the published version of the manuscript.

**Funding:** This insight was supported by the Natural Science Foundation of Guangdong Province (2022A1515010689), the Key Field Research and Development Program of Guangdong Province (2018B030337001), The National Natural Science Foundation of China (81830032, 31872779, and 82071421), the Guangzhou Key Research Program on Brain Science (202007030008).

**Data Availability Statement:** The datasets mentioned in this report can be found in online repositories. The names of the repository/repositories and accession number(s) can be found below: https://www.ncbi.nlm.nih.gov/bioproject/PRJNA688531.

**Conflicts of Interest:** The authors declare no conflict of interest.

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
