# Peer review of "Dynamic 5-Hydroxymethylcytosine Change: Implication for Aging of Non-Human Primate Brain"

_2075-4655, 2022_

Round 1

Reviewer 1 Report

The commentary by Liu et al. on the dynamic change of 5-hydroxymethylcytosine in non-human primate brains is a valuable contribution on our current understanding of the epigenetic changes during aging and development. The authors correctly point out the disadvantages of using rodent models versus non-human primates. Overall, I support the publication of this commentary. However, there are some typos and grammatical errors that should be corrected before publication. To name a few:

Page 1, line 35 in terms of (instead in term of)

Page 1, line 36 life span (instead of live span) and there other parts in this manuscript where lifespan is written, please decide on a uniform writing

Page 2, line 50 findings (instead of fundings)

References on page 3 (line 102 and 103 are referred to as ‘ref’), please correct.

Author Response

We thank the reviewer for the appreciation of our commentary. We also thank the reviewer for kindly pointing out some typos and grammatical issues. We have carefully checked the text and corrected grammar errors and marked them up using “Track Changes” in revised manuscript.

Reviewer 2 Report

The “methylation clock” was first introduced based on the tissue-specific DNA methylation (5mC) association with biological age. Because of the high epigenetic resemblance to humans, non-human primates including rhesus monkeys, are better animal models for ageing and neurodegenerative diseases. The authors summarise the work in which the enrichment and genome-wide distribution of  5hmC in the rhesus brain was profiled. The results suggest that 5hmC can be considered as a methylation clock along with 5mC, and the investigations of the functional roles of 5hmC in transcriptional regulation and ageing are warranted.

Thus this commentary is informative and relevant. And I am looking forward to more related exciting works from the authors.

Author Response

We thank the reviewer for the appreciation and recognition of our work. With more data coming along, we hope we can provide more perspectives on the significance of DNA methylation in aging and development in the near future.

Reviewer 3 Report

This is a very interesting manuscript that comments on new evidence about the tissue-specific pattern and dynamic accumulation of 5-hydroxymethylcytosine in the brain with aging. The authors focus on their own novel research findings on the profile of 5hmC at different ages using the rhesus monkey animal model and make a correlation with epigenetic profiling in the brain of other animal models and with human brain. The article is well written, logically constructed and concise. Below, there are a few minor suggestions that will improve readability.

L34: “Caenorhabditis elegans” instead of “elegans”.

L45: The font size of: "post mortem materials" seems to be higher than the rest the text.

L50: "fundings" should be "findings"

L101: The article of Schlosberg et al has already been published. Authors should consider citing the published article instead of the preprint.

Christopher E. Schlosberg, Dennis Y. Wu, Harrison W. Gabel and John R. Edwards. ME-Class2 reveals context dependent regulatory roles for 5-ydroxymethylcytosine. Nucleic Acids Research, 2019, Vol. 47, No. 5 e28. doi: 10.1093/nar/gkz001

Author Response

We thank the reviewer for the recognition and appreciation of our commentary. We also thank the reviewer for kindly pointing out some typos and citing issues. We have carefully checked the text and corrected these errors and marked them up using “Track Changes” in revised manuscript.